# Effects of ε-Poly-L-Lysine Combined with Wuyiencin as a Bio-Fungicide against *Botryris cinerea*

**DOI:** 10.3390/microorganisms10050971

**Published:** 2022-05-05

**Authors:** Zhaoyang Lv, Yanxuan Lu, Boya Li, Liming Shi, Kecheng Zhang, Beibei Ge

**Affiliations:** 1Institute of Plant Protection, Chinese Academy of Agricultural Sciences, Yuanmingyuan West Road, No. 2, Haidian District, Beijing 100193, China; lvcaas@163.com (Z.L.); lyx15673273360@126.com (Y.L.); l18703242721@163.com (B.L.); shiliming1983@163.com (L.S.); kczhang@ippcaas.cn (K.Z.); 2College of Forestry, Hebei Agricultural University, Lingyusi Street, No. 289, Baoding 071001, China

**Keywords:** ε-PL, wuyiencin, fungicide, mycelial growth, induced resistance

## Abstract

This study mainly evaluated the broad-spectrum fungicidal activity of ε-poly L lysine (ε-PL) against 12 pathogenic fungi. We further demonstrated synergistic antifungal activity of ε-PL combined with wuyiencin against *Botryris cinerea*. The combined bio-fungicide achieved an inhibition rate of 100% for mycelial growth using ε-PL at 500 μg/mL + wuyiencin at 50 μg/mL and for spore germination using ε-PL at 200 μg/mL + wuyiencin at 80 μg/mL in vitro. This synergistic spore and mycelia-damaging effect of the combination was confirmed using scanning electron microscopy. In vivo assays with combined bio-fungicide (1500 μg/mL ε-PL + 60 μg/mL wuyiencin) on detached leaves showed depressed growth and development of the spores of *B. cinerea*. The synergistic effect was further tested in combinations of ε-PL with wuyiencin by measuring the fractional inhibition concentration index (FICI) value below 0.5. Moreover, ε-PL and wuyiencin inoculation before *B. cinerea* infection significantly increased the superoxide dismutase, peroxidase, catalase, and phenylalanine ammonia-lyase activities, which suggested their involvement in tomato defense responses to disease to minimize damage to *B. cinerea.* These findings revealed that a combined bio-fungicide comprising ε-PL and wuyiencin had a good prospect for controlling plant fungal disease.

## 1. Introduction

Worldwide, crops are severely damaged by plant pathogens, causing reductions in the quantity and quality of crops and inducing significant economic losses, thereby posing a threat to global food security [1,2]. To manage plant diseases, a number of strategies have been used, including agriculture control, physical control, chemical control, and biological control [3,4]. The application of such methods has improved crop quality and production significantly in recent decades. With the increased focus on environmental and food safety, control methods have evolved to reduce the dependence on chemical pesticides, accompanied by the increased use of biocontrol strategies to manage plant diseases. Microbial antagonists play a significant role in controlling plant pathogens and diseases and are used as Biocontrol Agents (BCAs). Such microbial antagonists have been developed to control plant pathogenic diseases worldwide [2,5].

Fermentation of *Streptomyces* produces the natural antimicrobial, ε-Poly-L-lysine (ε-PL), a poly-cationic peptide comprising 25–30 L-lysine residues, which are connected via one amido bond between the ε-amino group and the α-carboxy group [6]. ε-PL shows marked antimicrobial activity against a variety of Gram-negative and Gram-positive bacteria, certain viruses, and fungi [7]. Furthermore, it has broad-spectrum antimicrobial activity, good water solubility, high-temperature stability, and is non-toxic to the human body, showing good biodegradability in humans and the environment. This non-toxicity and good antibacterial activity has led ε-PL to be generally recognized as safe as a food preservative for rice, vegetables, fruit, soft drinks, fish, and other products in many countries, including the United States, Korea, Japan, and China [8,9]. Mechanistically, ε-PL is believed to invoke electrostatic interactions that damage the cell membrane, ultimately disintegrating the microbial cells [10,11,12].

ε-PL is usually used as a biological antimicrobial in the post-harvest preservation of fruit and vegetables. However, some pathogenic infections infect crops and vegetables not only in post-harvest storage but also in before-harvest storage. In fact, because of its antibacterial activity and non-toxicity, ε-PL has great potential for controlling plant diseases in cultivation as an effective and environmentally safe biocontrol strategy.

Treatment with ε-PL in tobacco BY-2 protoplasts significantly downregulated *Tobacco Mosaic Virus* (TMV) RNA accumulation and induced the host defense response, suggesting ε-PL as a natural microbial to treat plant virus diseases [13]. In addition, ε-PL markedly inhibited *Alternaria alternata* disease progression by reducing germ tube elongation and spore germination and downregulating the expression of crucial fungal development-related genes [14]. In addition, the incidence of gray mold rot (caused by *Botryris cinerea*) on a variety of fruit and vegetables was effectively inhibited by ε-PL [15,16]. Taken together, these previous studies identified ε-PL as a potentially excellent biocontrol agent with direct antifungal activity. Moreover, ε-PL combined with chitooligosaccharide showed a synergistic effect in controlling gray mold rot in tomatoes, i.e., the combination induced an increased inhibition rate compared with that of ε-PL alone [16]. Hence, it would also be significant to screen biocontrol components that have synergistic effects with ε-PL. Wuyiencin is produced by *Streptomyces albulus* CK-15 and is widely used as an antifungal agent to control various fungal diseases of vegetables and field crops in agriculture [17]. However, there have been few studies conducted on ε-PL combined with other second metabolites agents from *Streptomyces* to inhibit plant diseases.

Herein, we aimed to test the antifungal activity of ε-PL combined with wuyiencin as bio-fungicides against 12 species of pathogenic fungi. We revealed that ε-PL combined with wuyiencin had synergistic effects against *B. cinerea* by suppressing mycelial growth and spore germination in vitro and in vivo; it induced plant disease resistance, as demonstrated by defense-related enzyme activity assays.

## 2. Material and Methods

### 2.1. ε-PL and Wuyiencin Production and Assays

*Streptomyces albulus* CK-15, which was provided by the China General Microbiological Culture Collection Centre (CGMCC no. 0703), was used to prepare ε-PL. Briefly, Sephadex G-25 column chromatography was used to purify ε-PL from *S. albulus* CK-15 fermentation broth. Then, matrix-assisted laser desorption ionization time-of-flight mass spectrometry (MALDI-TOF-MS) and high-resolution magic angle spinning nuclear magnetic resonance (HR-MAS NMR) were used to characterize the purified ε-PL. The ε-PL preparation was over 99% pure and contained approximately 30 residues. The crude powder of wuyiencin was extracted from the fermentation broth. Wuyiencin was separated and purified by Sephadex G-25 and ODS column chromatography and semipreparative high-performance liquid chromatography. The wuyiencin preparation was over 95% pure.

### 2.2. Pathogenic Fungi and Plants

A total of 12 pathogenic fungi were used: *Physalospora piricola*, *Rhizopus stolonifer*, *Botrytis cinerea*, *Fulvia fulva*, *Sclerotinia sclerotiorum*, *Gibberella sanbinetti*, *Colletotrichum lagenarium*, *Phytophthora infestans*, *Valsa mali*, *Alteraria alternata*, *Helminthosporium maydis*, and *Phomopsis asparagi*, which were stored in the Institute of Plant Protection, Chinese Academy of Agriculture Science, Beijing, China. These fungi were stored at 4 °C on potato dextrose agar (PDA) medium. Prior to the antagonist experiments, the fungi were activated at 25 °C for 5–7 days to determine their mycelial growth rates. Tomato plants were grown in a 1:1 mixture of nutritional soil and sand, by volume, in 20 cm-diameter plastic pots and incubated under greenhouse conditions at 25 °C, 90% relative humidity, and a 12 h light and 12 h dark photoperiod.

### 2.3. Effect of ε-PL on Mycelial Growth 

The antifungal activity of ε-PL toward the 12 pathogens listed in Section 2.2 was assayed in vitro by determining their mycelial growth. ε-PL solution was added to PDA medium at final concentrations of 100, 300, 500, 1000, and 2000 parts per million (ppm); the control comprised PDA lacking ε-PL solution. We placed the 6 mm diameter mycelial plugs in the center of the PDA plate, followed by culture at 25 °C in the dark. At 5 days post-inoculation (dpi), the mean diameters of the colonies were determined. Each pathogen had three inoculation replicates, and we performed the growth assays independently three times. The rate of inhibition of mycelial growth by ε-PL was calculated using the following formula: Inhibition rate (%) = [(control diameter-treated diameter)/control diameter] × 100%. The 50% effective concentration (EC_50_) values of the agent toward the pathogens were calculated according to the linear regression of the colony diameters for log10-transformed standard concentrations, repeated 3 times.

### 2.4. Synergistic Effects of ε-PL Combined with Wuyiencin on B. cinerea Mycelial Growth 

The synergistic effects of ε-PL combined with wuyiencin on *B. cinerea* mycelial growth were assessed using the radial growth test on PDA. The ε-PL solution at 5 different concentrations (100, 200, 300, 400, and 500 μg/mL) and wuyiencin solution at 3 different concentrations (12.5, 25, and 50 μg/mL) were tested. Different solutions were mixed with PDA medium, and medium lacking both ε-PL and wuyiencin solution was used as a control. Mycelial growth was assayed as in Section 2.3
*B. cinerea* inoculation was replicated three times, and the inhibition rate was calculated as in Section 2.3.

### 2.5. The Effects of Treatment on B. cinerea Mycelial Morphology 

*B. cinerea* mycelia sampled from the edge of 5 day-old colonies were added to plates lined with cellophane containing PDA medium with ε-PL at 500 μg/mL, wuyiencin at 25 μg/mL, and ε-PL at 500 μg/mL + wuyiencin at 25 μg/mL, respectively. Plates without any treatment served as controls. The plates were incubated at 25 °C in the dark for 72 h. Thereafter, we observed the morphology of the top layer of the fungus under light microscopy (Olympus IX-71, Tatsuno, Japan). The experiment was carried out independently three times.

### 2.6. Detection of the Spore Germination of B. cinerea

To evaluate the effect of ε-PL, wuyiencin, and their combination on spore germination, *B. cinerea* was cultured at 25 °C in an incubator for 7 days until the fungus had grown to completely fill the dish. We made a spore suspension, which was subjected to filtration to achieve a concentration of 50–60 conidia per microscopic field (magnification = 10×). ε-PL was prepared at 50, 100, 200, 300, 400, and 500 μg/mL. Wuyiencin was diluted to 10, 20, 40, 60, 80, and 120 μg/mL. In addition, combination treatments were as follows: ε-PL 200 μg/mL + wuyiencin 10 μg/mL, ε-PL 200 μg/mL + wuyiencin 20 μg/mL, ε-PL 200 μg/mL + wuyiencin 40 μg/mL, ε-PL 200 μg/mL + wuyiencin 60 μg/mL, ε-PL 200 μg/mL + wuyiencin 80 μg/mL, and ε-PL 200 μg/mL + wuyiencin 120 μg/mL. A total of five milliliters of the *B. cinerea* spore suspension was added to the different concentrations of ε-PL, wuyiencin, and their combinations. Then, five milliliters of the *B. cinerea* spore suspension were added to distilled water as a control, and each treatment was repeated three times. Samples (20 μL) of the different treatments were cultured on slides with absorbent paper moistened with distilled water in a 24 °C constant temperature incubator, and the spore germination of the samples was observed every day until spore germination was over 90%. The spore germination rate was calculated for ≥200 spores each time. The experiment was performed three times.
Germination rate (%) = Number of germinated spores/Total number of spores × 100%
Inhibition rate (%) = (Spore germination rate of the control group − Spore germination rate of treatment group)/Spore germination rate of the control × 100%

### 2.7. Antifungal Activity on Detached Leaves

Tomato seeds were grown under the conditions of a 12 h light–dark cycle at 25 °C. When the tomato seedlings grew to the 5–6 leaf stage, leaves with consistent growth were selected, washed with sterile water to remove surface impurities, and the petiole was covered with cotton to retain moisture. We sprayed the leaves with various ε-PL solutions: 1500, 2000, 2500, and 3000 μg/mL; wuyiencin solution (60 μg/mL); and various combinations: wuyiencin 60 μg/Ml + ε-PL 1500 μg/mL, wuyiencin 60 μg/mL + ε-PL 2000 μg/mL, wuyiencin 60 μg/mL + ε-PL 2500 μg/mL and wuyiencin 60 μg/mL + ε-PL 3000 μg/mL. Control leaves were sprayed using distilled water. Then, 24 h later, the leaves were sprayed with a 2% glucose solution, inoculated with *B. cinerea*, and cultured in an incubator with 95% humidity and a 12 h light and 12 h dark photoperiod with daily water spraying to retain moisture. After 3 days, the diameter of the leaf lesion formed around each inoculated mycelial plug was measured, and the percent inhibition of each treatment was calculated. A total of 20 leaves were inoculated with each concentration. The assays were carried out three times independently.

Percent inhibition (PI) = (dc − dt)/dc × 100%, where dc represents the diameter of the control (untreated) colony and dt represents the diameter of the treated colonies.

### 2.8. Effects of the Combined Treatment on the Pathogen on Isolated Tomato Leaves under a Scanning Electron Microscope

*B. cinerea*-inoculated tomato leaves were sprayed with distilled water (control), ε-PL at 1500 μg/mL, wuyiencin at 60 μg/mL, and ε-PL at 1500 μg/mL + wuyiencin at 60 μg/mL), respectively. The leaves were collected at 72 h post-inoculation and fixed using glutaraldehyde following the standard protocol [18]. Leaf samples were mounted onto aluminum stubs and spatter-coated using Au–Pd. A JEOL JEM 5410 Scanning electron microscope (JEOL, Tokyo, Japan) at 20 kV was used to observe the samples for the effects of the treatments on *B. cinerea* morphology and ultrastructure.

### 2.9. Determination of Synergistic Effect of ε-PL Combined with Wuyiencin

The antimicrobial combinations of wuyiencin and ε-PL were determined using the method of Itamar Shalit et al. [19] with slight modification. Drug interactions were assessed by checkerboard assays using the NCCLS M38-P microdilution methodology in standard 96-well sterile flat-bottom polystyrene plates (Corning). The species of selected fungus was *B.cinerea*. The concentration of *B.cinerea* spores was adjusted to 1.0 × 10^4^ CFU/mL by hemocytometer. The concentration of wuyiencin was diluted to 0–80 μg/mL and ε-PL to 0–100 μg/mL in PDB medium. Each well added 100 μL of the diluted drug concentrations and 100 μL of conidia (final volume of each well, conidia and drugs, 200 μL). The wells of 100 μL conidia + 100 μL media and 100 μL sterile water + 100 μL media were set up as controls. The plate was scored at 23 °C for 72 h. Optical density readings were performed at a wavelength of 600 nm after 72 h and were preceded with an automated 120 s shaking to ensure uniform distribution of well contents. The assays were carried out three times independently.

To assess possible synergistic activity, the fractional inhibitory concentration. FICI was calculated as follows:FICI = FIC_ε-PL_ + FIC _wuyiencin_ = (ε-PL)/MIC_ε-PL_ + (wuyiencin)/MIC_wuyiencin_
where the MIC was the lowest drug concentration resulting in complete inhibition of hyphal growth. (ε-PL) and (wuyiencin) indicate the MIC of ε-PL and wuyiencin used for complete inhibition in combination; MIC_ε-PL_ and MIC_wuyiencin_ indicate the minimum inhibitory concentrations of ε-PL and wuyiencin when used alone.

FICI values were interpreted as follows: FICI < 0.5, synergistic; 0.5 < FICI < 1, synergistic to additive; 1 < FICI < 4, indifferent; FICI > 4, antagonistic.

### 2.10. Treatment Responses of the Tomato Antioxidant Enzyme System

To assess the responses of the plant antioxidant enzyme system, different bio-fungicides and distilled water were applied 3 days prior to *B. cinerea* spores inoculation. We compared tomato plants after treatment using distilled water (control), wuyiencin at 60 μg/mL, ε-PL at 1500 μg/mL, and wuyiencin at 60 µg/mL + ε-Pl at 1500 µg/mL. The leaves were collected at 0, 6, 12, 24, and 48 h post-treatment. One gram of leaf sample was ground in liquid nitrogen, homogenized in 5 mL of 0.02 mol/L phosphate buffer (pH 6.8) at 0–4 °C, and centrifuged at 8000× *g* at 0–4 °C for 20 min. The supernatant comprising the enzyme solution was stored at 4 °C for subsequent analyses.

#### 2.10.1. Catalase (CAT) Activity Determination

CAT activity was determined using the method of García et al. [20]. with slight modifications (e.g., OD_240_).

#### 2.10.2. Superoxide Dismutase (SOD) Activity Determination

SOD activity was determined using the method of Pokora et al. [21], with slight modifications (e.g., 0.3 mL of 0.1 mmol/L EDTA-Na_2_ and 0.3 mL of 0.02 mmol/L riboflavin; pH 7.8).

#### 2.10.3. Detection of Peroxidase (POD) Activity Determination

POD activity was determined using the method described by Wang et al. [22], with slight modifications.

#### 2.10.4. Phenylalanine Ammonia Lyase (PAL) Activity Detection

PAL activity was determined using the method described by Li et al. [23], with slight modifications (i.e., absorption at OD_290_).

### 2.11. Statistical Analysis

Analysis of variance (ANOVA) in SPSS 13.0 (IBM Corp., Armonk, NY, USA) was used for the statistical analyses of the data. Fisher’s protected least significant difference (LSD) tests were used to compare the significant differences between the means. *p* < 0.05 was considered to indicate a significant difference.

## 3. Results

### 3.1. The Spectrum of ε-PL’s Antifungal Activity

Selected plant pathogenic fungi (*Physalospora piricola*, *Rhizopus stolonifer*, *Botrytis cinerea*, *Fulvia fulva*, *Sclerotinia sclerotiorum*, *Gibberella sanbinetti*, *Colletotrichum lagenarium*, *Phytophthora infestans*, *Valsa mali*, *Alteraria alternata*, *Helminthosporium maydis*, *Phomopsis asparagi*) were used to estimate ε-PL’s antifungal activity. ε-PL displayed an obvious inhibitory effect on the mycelia growth of various pathogenic fungi in a concentration-dependent manner (Figure 1). *Physalospora piricola*, *Rhizopus stolonifer*, *Botrytis cinerea*, *Valsa mali*, and *Helminthosporium maydis* could not grow at all under the high concentration of ε-PL, 2000 μg/mL, with inhibition rates of 100% (Table 1). With the lower concentration of 500 μg/mL ε-PL, the inhibition rate against *B. cinerea*, *Colletotrichum lagenarium*, *Valsa mali*, and *Alteraria alternata* was above 70%. Data analysis of the EC_50_ was carried out based on the inhibition rate, and the EC_50_ values were all less than 400 μg/mL, respectively (*p* < 0.05) (Table 1).

### 3.2. Effect of Combined Treatment on the Mycelial Growth of B. cinerea

ε-PL could inhibit the growth of *B. cinerea* significantly; therefore, we tested the inhibitory effect of ε-PL combined with wuyiencin. The inhibitory effect increased as the concentrations of ε-PL and wuyiencin increased. The inhibition rate was markedly higher following the combined treatment compared with ε-PL treatment alone (Figure 2). The inhibition rate of the highest concentration of ε-PL, 500 μg/mL, when used alone was 71.15% and the inhibition rate of wuyiencin at 25 μg/mL, when used alone, was 38.81% (Table 2). When the 2 compounds were used together at these concentrations, a significantly increased inhibition rate of 87.71% was achieved (Appendix A). Moreover, the mycelial inhibition rate was up to 100% when the fungus was treated with ε-PL 500 μg/mL + wuyiencin 50 μg/mL.

### 3.3. Effect of Treatment on the Mycelial Morphology of B. cinerea 

Based on the above results, 3 treatments (ε-PL 500 μg/mL, wuyiencin 25 μg/mL, and ε-PL 500 μg/mL + wuyiencin 25 μg/mL) were selected to further observe the effects of treatment on *B. cinerea* mycelial morphology. Distilled water treatment was used as a control. The mycelia of the control group were slender and straight, with uniform thickness, smooth lines, and normal branches. The formation of branches was at a certain distance from the top, and the cytoplasm was uniform and transparent (Figure 3a). The mycelia treated with wuyiencin showed less branching, and the tip of the growth point was enlarged with uniform and transparent cytoplasm (Figure 3c). The mycelia treated with ε-PL showed disorganized growth, increased branches, shorter branch spacing, shorter terminal branches, and uniform and transparent cytoplasm (Figure 3b). The mycelia treated by wuyiencin + ε-PL showed disorganized growth, with a large number of branches at the growing point, shortened branch spacing and terminal branches, the top of the growing point was expanded, the mycelia no longer displayed obvious growth, and the cytoplasm appeared coagulated (Figure 3d).

### 3.4. Detection of B. cinerea Spore Germination

ε-PL and wuyiencin could effectively inhibit conidial germination of *B. cinerea*, and the inhibition rate increased with increasing concentration. When ε-PL was used alone, the conidial inhibition rate reached 58.82% at 200 μg/mL, and when wuyiencin was used alone, the conidial inhibition rate reached 76.47% at 60 μg/mL. When 60 μg/mL wuyiencin and 200 μg/mL ε-PL were combined, the conidial inhibition rate reached 100% (Table 3). The microscopic observation after 36 h of treatment showed that very few spores could produce bud tubes, which were shrunken and deformed after combined treatment compared with those in the distilled water control group (Figure 4). 

### 3.5. Antifungal Activity on Detached Leaves

Detached tomato leaves were used to determine the inhibitory effect of the combination treatment on *B. cinerea* infection. The antifungal activity on detached leaves was different from that in solid media, i.e., the inhibition rate increased with increasing concentrations of treatment. By contrast, with increasing ε-PL concentration, the control effect decreased and the leaves appeared chlorotic and had black spots (Figure 5). When ε-PL at 1500 μg/mL was used alone, the inhibition rate was the highest at 72.22% (Table 4). When wuyiencin was used alone at 60 μg/mL, the inhibition rate was only 38.89%. The addition of wuyiencin to each concentration of ε-PL increased the inhibition rate of all combined treatments, for example, the inhibition rate ranged from 72.22% (ε-PL 1500 μg/mL) to 88.89% (ε-PL 1500 μg/mL + wuyiencin 60 μg/mL) and from 61.67% (ε-PL 2000 μg/mL) to 83.33% (ε-PL 2000 μg/Ml + wuyiencin 60 μg/mL) (Table 4). These results allowed us to conclude that the addition of wuyiencin could improve the inhibition rate and reduce the harm caused by high concentrations of ε-PL.

### 3.6. Scanning Electron Microscope Observation of the Effects of Combined Treatment on the Pathogen on Isolated Tomato Leaves 

Compared with the control group (treated with distilled water), *B. cinerea* treated with ε-PL at 1500 μg/mL, wuyiencin at 60 μg/mL, and their combination (ε-PL at 1500 μg/mL + wuyiencin at 60 μg/mL) showed changes in the quantity, morphology, and spore production of mycelial (Figure 6). Treatment with the combination therapy resulted in the lowest of mycelia density (Figure 6d), followed by those after treatment with 60 μg/mL wuyiencin and 1500 μg/mL ε-PL alone (Figure 6b,c).

The mycelia in the control group were well stretched, smooth, and full; uniform in thickness; smooth in line; and tapered at the growth point (Figure 6e). The mycelia treated with 60 μg/mL wuyiencin were slightly distorted, dry and fine; the branches spacing became shorter; and the growth point was malformed (Figure 6f). In the 1500 μg/mL ε-PL treatment group, the mycelia were distorted and deformed; some mycelia collapsed; and the top of the growth point was blunt and round (Figure 6g). In the combination treatment group, the mycelia were severely deformed and slimmer; some mycelia were twisted into a spiral shape; and the growth point was deformed, being enlarged or candidial-shaped (Figure 6h). The control group produced the most spores, and the spores were smooth, plump, and well developed (Figure 6i). After treatment with wuyiencin or ε-PL alone, the spore yield decreased and the spores contracted slightly (Figure 6j,k). After the combined treatment, the spore yield decreased significantly, the spores germination were depressed and deformed, and the growth and development of the spores were inhibited such that they could not germinate normally (Figure 6l).

### 3.7. Determination of Synergistic Effect of ε-PL Combined with Wuyiencin

The FICI method was chosen to determine the combined action of ε-PL and wuyiencin, since it is widely used in synergy studies. The FICI values obtained for each test isolate at 72 h are shown in Table 5. The results showed that the combination of MIC of wuyiencin and ε-PL were 5 μg/mL and 10 μg/mL, which is far below the MIC when used alone (wuyiencin 40 μg/mL and ε-PL 30μg/mL) in three duplicates. The FICI value of ε-PL combined with wuyiencin aganist *B.cinerea* was 0.46 < 0.5, showing synergistic effect, due to the fact that synergy is attributed to any interaction with the FIC below 0.5. A promising combination of ε-PL and wuyiencin was considered to be used as biological fungicide products because of their powerful synergy against *B.cinerea.*

### 3.8. Defense-Related Enzyme Activities in Tomato

Considering the very efficient inhibition by the combined treatment (60 μg/mL wuyiencin + 1500 μg/mL ε-PL), we speculated the addition of wuyiencin would activate systemic resistance-related enzymes (CAT, POD, SOD and PAL); therefore, the activities of these antioxidant enzyme activity were examined.

The CAT activity increased initially, peaked at 12 h, and decreased in the infected leaves in the control and treatment groups. The CAT activity was highest in combined treatment group, being 2.58 times higher than that in the control group at 24 h (Figure 7a).

The same trend occurred for POD activity in the control group and single 60 µg/mL wuyiencin treatment, both of which showed gradually increased activity. In the single ε-PL and combined treatment (1500 µg/mL ε-PL + 60 µg/mL wuyiencin) the POD activity showed a similar gradual increase in all treatments up to 12 h, peaking at 12 h, and then declining. In the combined treatment group, POD activity was highest at 12 h, being 3.66 times the activity in the control group (Figure 7b). This increased POD activity was probably a response to the excess H_2_O_2_ production by the increased SOD activity in the combined treatment group.

The treatments resulted in different increases in SOD activity after 0 h compared with that in the control. In the control and treatment groups, the SOD activity in the infected leaves increased rapidly within 12 h and then decreased. At 24 h, the SOD activity was slightly higher in the ε-PL and wuyiencin single treatment groups compared with those in the control group. Interestingly, in the combined treatment group, the SOD activity in the infected leaves increased suddenly and reached a maximum during 24–48 h. The SOD activity was significantly higher than that in the control group (Figure 7c).

The PAL activity of plants treated with 1500 µg/mL ε-pL and 60 µg/mL wuyiencin increased gradually within 48 h. After the combined treatment, the PAL activity of the inoculated plants reached a peak at 12 h, which was 1.35, 1.23, and 1.22 times that of the control, 60 µg/mL wuyiencin treatment, and 1500 µg/mL ε-PL treatment, respectively. Thus, treatment with 60 µg/mL wuyiencin and 1500 µg/mL ε-PL could increase the PAL activity in plants, while combination treatment (1500 µg/mL ε-PL + 60 µg/mL wuyiencin) treatment could further increase the PAL activity in plants, thus making tomato plants more resistant to *B. cinerea* (Figure 7d).

## 4. Discussion

ε-PL has been used widely and is recognized as a safe food preservative [11]. ε-PL has strong antibacterial, antiviral, and antifungal activities [24,25]; therefore, it has been gradually applied in the field of plant disease control as a natural anti-microbial product in agriculture in recent years. ε-PL has been reported to inhibit the growth of fungal pathogens as an antifungal activity agent, inhibiting the mycelial growth and spore germination of *Sclerotinia scleroriorum*, *B. cinerea*, and *Alternaria alternate* [13,26,27] which was consistent with our results showing effective inhibition of the mycelial growth of *Botrytis cinerea* and *Alternaria alternata* by ε-PL (Table 1). It is worth noting that ε-PL at 200 µg/mL had an inhibitory rate toward mycelial growth of 83.9% against *A. alternata* [13]. It is quite different from what was observed in our experiment, which showed ε-PL 300 µg/mL had an inhibition rate of 26.47% against *A. alternata*. This difference is likely due to the genetic sensitivity and resistance to different sources of *A. alternata* strains. Meanwhile, ε-PL at the same concentration or lower did inhibit the mycelial growth of *Sclerotinia scleroriorum* and *B. cinerea.* ε-PL at 1200 µg/mL reduced the lesion by 76.28% and also reduced the necrotic lesion area significantly at 600–1500 µg/mL [27]. Our results also demonstrated broad-spectrum effects against 12 pathogenic fungi. ε-PL had a significant inhibitory effect on *Physalospora piricola*, *Rhizopus stolonifer*, *Botrytis cinerea*, *Valsa mali*, and *Helminthosporium maydis*. These pathogens could not grow at all (100% inhibition) when the concentration of ε-PL was 2000 µg/mL. Thus, we concluded that ε-PL as a single agent could inhibit infection by plant fungal pathogens.

However, it has been reported that ε-PL is deleted rapidly after its initial application and quickly loses its activity [28]. In this case, combining ε-PL with other agents is a good strategy to reduce the ε-PL dose while maintaining the inhibitory effect. ε-PL combined with chitooligosaccharide (COS; 200 µg/mL ε-PL + 400 µg/mL COS) displayed stronger inhibitory activity (90.22%) against *B. cinerea* compared with treatment using either agent alone [16]. In addition, combinations of ε-PL with other compounds, such as endolysin, chitosan, and nisin, demonstrated synergistic antibacterial effects against different pathogens [6,29,30,31]. The *S. albulus* CK-15 secondary metabolite wuyiencin is used widely in agriculture to control fungal diseases, including tomato gray mold, strawberry powdery mildew, and cucumber downy mildew [32]. Wuyiencin markedly downregulates mycelial growth and spore formation/germination and disturbs mycelium membrane permeability during primary pathogen infection, which additionally induces host plant defense reactions. Moreover, it downregulated the expression of *Sclerotinia sclerotiorum* growth and infection-related genes [17]. Herein, we tested the inhibitory effects ε-PL combined with wuyiencin. The combined treatment significantly inhibited the mycelial growth and spore germination of *B. cinerea*. The inhibition of *B. cinerea* mycelial growth using ε-PL at 500 μg/mL and wuyiencin at 50 μg/mL alone were 71.15% and 88.10%, respectively. Combined treatment using ε-PL concentrations ranging from 200 to 500 μg/mL and wuyiencin at 50 μg/mL resulted in the inhibition rate ranging from 95.21% to 100% (Table 2). Under these treatments, the mycelia no longer showed obvious growth, and the cytoplasm appeared coagulated (Figure 3d). Similarly, combined treatment with ε-PL at 200 μg/mL + wuyiencin at 60 μg/mL completely inhibited spore germination (Figure 4). In vitro, the growth of tomato gray mold was inhibited by 72.22% by 1500 μg/mL ε-PL (Table 4).

On the one hand, ε-PL could significantly reduce tomato gray mold, and on the other hand, it could induce resistance after a third spraying [16,26]. Reactive oxygen species (ROS) frequently build up in plants as a result of pathogen infection, and ROS have a well-established and pivotal role in plant defense response regulation [33,34]. However, plant cell death can be caused by high levels of ROS [35,36]. By contrast, low ROS levels could enhance tolerance to various types of stress [37,38]. Thus, there must be a balance between the production and scavenging of ROS if they are to be used as signaling molecules [39]. ROS-scavenging enzymes, such as POD, SOD, and CAT, have important functions in cellular ROS homeostasis. In the present study, the analysis of SOD, POD, and CAT activities showed increases in response to the combined treatment or single-agent treatment; however, the enzyme activities were significantly higher after combined treatment. The results indicated that the combined treatment could reduce ROS overaccumulation in response to *B. cinerea* infection and improve plant resistance to *B. cinerea* infection through the maintenance of balanced ROS levels.

PAL is the key enzyme in the phenylpropanoid pathway, regulating phenols, phytoalexins, and lignin synthesis, which are related to localized resistance processes [40]. PAL is activated when plants are infected by a pathogen [41], which was consistent with our results that PAL activity increased in all treatment groups after inoculation with *B. cinerea* (Figure 7d). Furthermore, the PAL activity was higher after combined treatment than after treatment with either agent alone. Increased levels of PAL have been shown to be correlated with resistance reactions [42]. Thangavelu et al. reported increased PAL activity in the roots and leaves of banana plants treated with *T. harzianum* [43]. Christopher et al. [44] also reported the induction of PAL activity in tomato plants by *Trichoderma virens* and *Fusarium oxysporum*. Therefore, the combined treatment could improve tomato disease resistance by increasing PAL activity.

In this study, we observed increased defense-related enzyme activity after treatment with the combination of wuyiencin and ε-PL lysine compared with that induced by either agent alone. Previous in vitro leaf tests showed that high concentrations of ε-PL had certain harmful effects on tomato plants. The harmful leaves appeared chlorotic, black spots were less resistant to plant pathogens, and the control effect decreased. However, when mixed with wuyiencin, not only did the prevention effect improve but also the damage was significantly reduced (Figure 4), which is consistent with the results of previous field trials of ε-PL on strawberries (Appendix A). In addition, wuyiencin has been reported to protect plants by triggering a defense response [17], which partially explains why the combination of wuyiencin and ε-PL in this study could reduce the harm caused by ε-PL alone in plants.

In conclusion, ε-PL had broad-spectrum fungicidal activity against various pathogenic fungi, reaching 100% for some fungi. This combined bio-fungicide (500 μg/mL ε-PL + 50 μg/mL wuyiencin) showed 100% inhibition of *B. cinerea* mycelial growth on solid media in vitro, with associated disturbed mycelial morphology. The concentration most suitable for inhibiting spore germination was 200 μg/mL ε-PL + 60 μg/mL wuyiencin. Meanwhile, in detached leaves with more complex environmental conditions, such as temperature and humidity, the combined bio-fungicide (1500 μg/mL ε-PL + 60 μg/mL wuyiencin) suppressed the normal growth and development of *B. cinerea* spores. Moreover, the combination bio-fungicide induced *B. cinerea* resistance in tomatoes. The combined bio-fungicide comprising ε-PL and wuyiencin could be developed for use in crop protection. However, whether the combined treatment can control *B. cinerea* infection of tomato plants requires field tests. There were also many aspects for further study, including compatibility of the bio-fungicides, the feasibility of the industrial scale-up, improvement of the rain fastness, evaluation of possible phytotoxicity issues, and efficacy tests using pathosystems closer to real agricultural practice starting from experiments under greenhouse conditions.

## Figures and Tables

**Figure 1 microorganisms-10-00971-f001:**
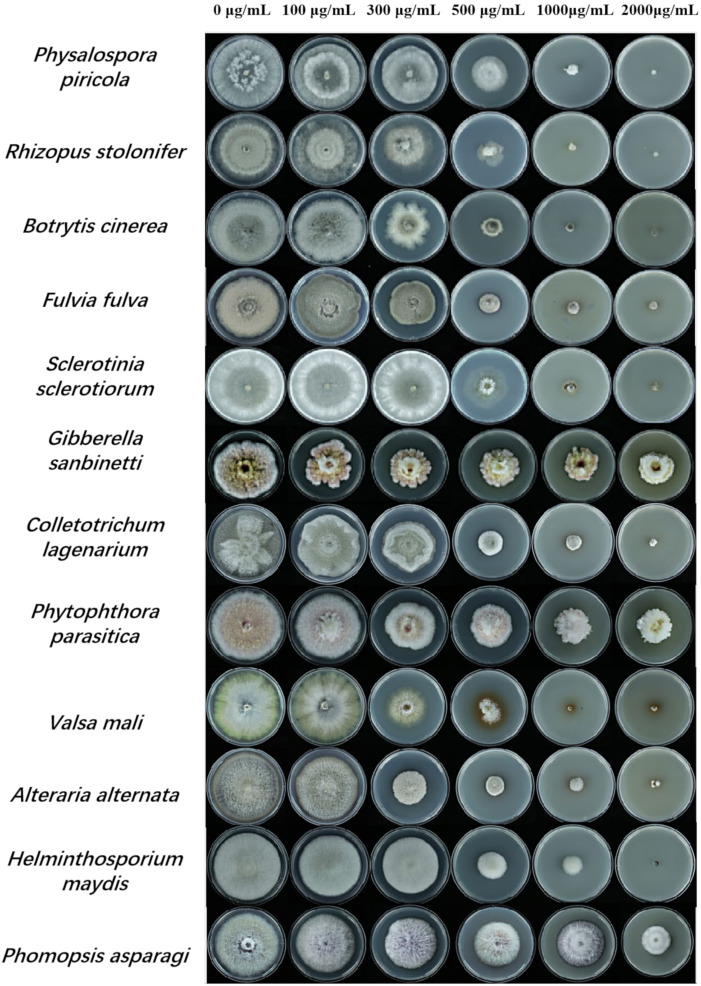
Antifungal spectrum of ε poly L lysine (ε-PL) treatment for different plant pathogenic fungi. When the fungus had grown to completely fill the plate in the control for each species (distilled water-treated), the colony diameters on the experimental plates were measured.

**Figure 2 microorganisms-10-00971-f002:**
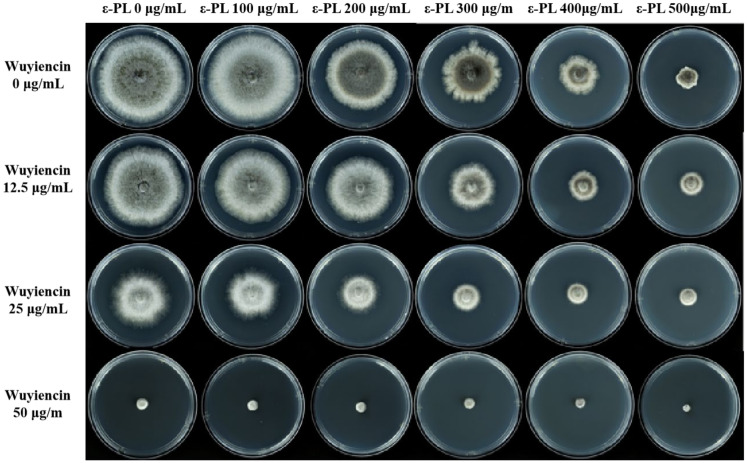
Inhibitory effect of different concentrations of wuyiencin + ε-PL on the mycelial growth of *Botrytis cinerea*.

**Figure 3 microorganisms-10-00971-f003:**
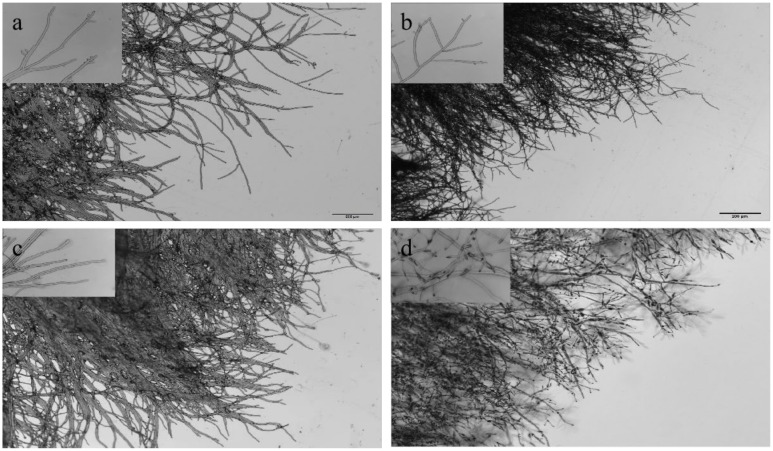
Effects of wuyiencin and ε-PL on the mycelial morphology of *Botrytis cinerea* under a microscope at 10× magnification. (**a**): Morphology after 72 h of treatment with distilled water. (**b**): Morphology after 72 h of treatment with 500 μg/mL ε-PL. (**c**): Morphology after 72 h of treatment with 25 μg/mL wuyiencin. (**d**): Morphology after 72 h of treatment with 500 μg/mL ε-PL + 25 μg/mL wuyiencin.

**Figure 4 microorganisms-10-00971-f004:**
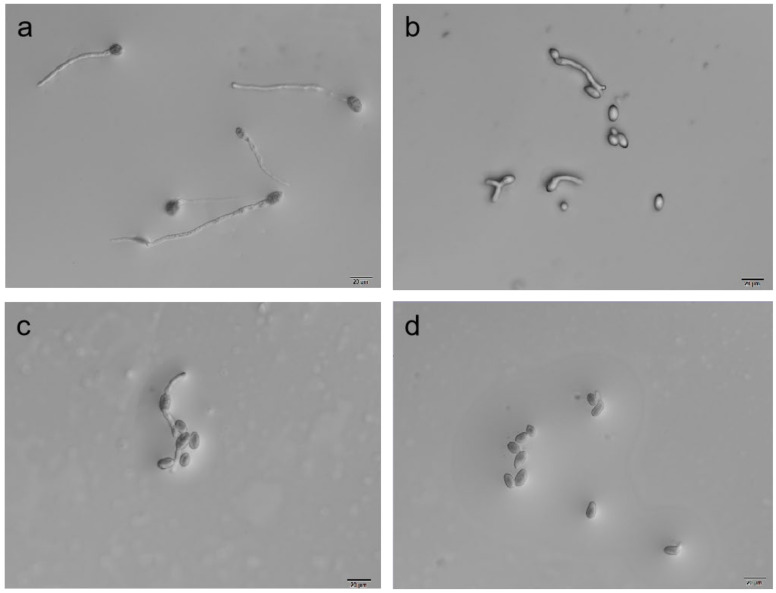
Effects of wuyiencin and ε-PL on conidial germination of *Botrytis cinerea* under a microscope at 40× magnification. (**a**): Germination at 36 h after treatment with distilled water. (**b**): Germination at 36 h after treatment with 200 μg/mL ε-PL. (**c**): Germination at 36 h after treatment with 60 μg/mL wuyiencin. (**d**): Germination at 36 h after treatment with 200 μg/mL ε-PL + 60 μg/mL wuyiencin.

**Figure 5 microorganisms-10-00971-f005:**
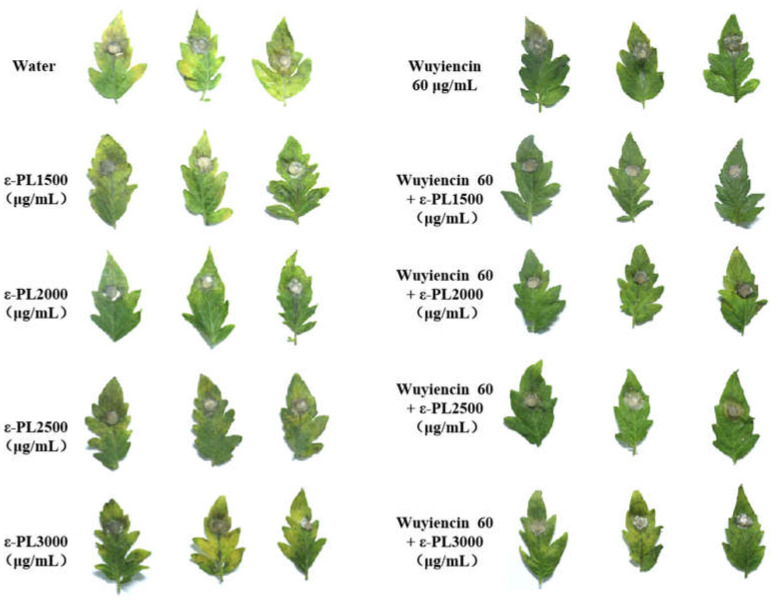
In vitro protective capability ε-PL combined with wuyiencin to control *Botrytis cinerea* on tomato leaves.

**Figure 6 microorganisms-10-00971-f006:**
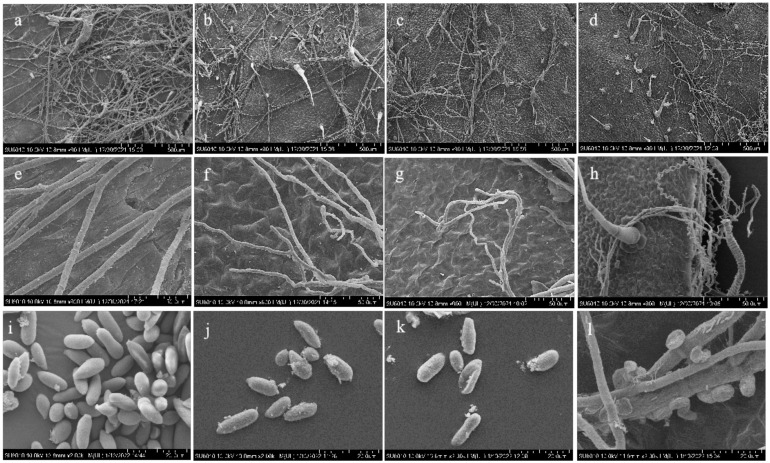
Effects of wuyiencin and ε-PL on the filaments and conidia of *B. cinerea* on tomato leaves. (**a**–**d**): The density of mycelia treated with distilled water, wuyiencin at 60 μg/mL, ε-PL at 1500 μg/mL, wuyiencin at 60 μg/mL + ε-PL at 1500 μg/mL for 72 h. (**e**–**h**): The mycelial morphology after the same treatments as in (**a**–**d**). (**i**–**l**): The morphology of conidia after the same treatments as in (**a**–**d**).

**Figure 7 microorganisms-10-00971-f007:**
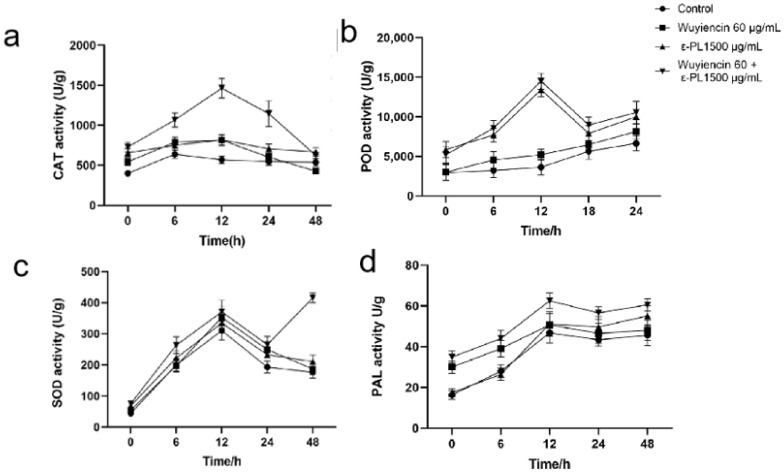
The effect of wuyiencin and ε-PL on defense-related enzyme activities in tomatoes. A total of 60 mg/L wuyiencin, 1500 μg/mL ε-PL, 60 mg/L wuyiencin + 1500 μg/mL ε-PL, and sterile water (control) were applied 3 days before the inoculation of *B. cinerea* spores (10^7^ cfu/mL). The treated tomato leaves were incubated at 25 °C with 95% relative humidity for 0, 6, 12, 24, and 48 h. Extracts were made at each time point for enzyme activity testing. (**a**): Catalase (CAT) activity changes in tomato leaves receiving the different treatments. (**b**): Peroxidase (POD) activity changes in tomato leaves receiving the different treatments. (**c**): Superoxide dismutase (SOD) activity changes in tomato leaves receiving the different treatments. (**d**): Phenylalanine ammonia-lyase (PAL) activity changes in tomato leaves receiving the different treatments.

**Table 1 microorganisms-10-00971-t001:** Antifungal spectrum of ε-PL treatment toward different plant pathogenic fungi.

Pathogenic Fungi	Concertation of ε-PL (µg/mL)	Inhibition Rate (%)	Toxicity Equation	Correlation Coefficient	EC_50_ (µg/mL)
*Physalospora piricola*	100	1.92 ± 0.33 e	y = 4.2465x − 6.1396	R^2^ = 0.9115	419.99
	300	16.35 ± 2.07 d			
	500	45.4 ± 3.18 c			
	1000	83.72 ± 6.29 b			
	2000	100 ± 0.00 a			
*Rhizopus stolonifer*	100	0.63 ± 0.08 e	y = 4.6286x − 7.1749	R^2^ = 0.9586	426.94
	300	16.98 ± 2.34 d			
	500	50.44 ± 6.29 c			
	1000	89.31 ± 5.78 b			
	2000	100 ± 0.00 a			
*Botrytis* *cinerea*	100	3.44 ± 0.47 d	y = 3.9943x − 5.0512	R^2^ = 0.9393	328.39
	300	36.5 ± 2.83 c			
	500	80.38 ± 5.48 b			
	1000	88.13 ± 7.29 b			
	2000	100 ± 0.00 a			
*Fulvia fulva*	100	3.78 ± 0.63 e	y = 2.7349x − 2.0522	R^2^ = 0.9789	378.96
	300	48.14 ± 3.76 d			
	500	69.76 ± 5.29 c			
	1000	83.51 ± 6.68 b			
	2000	97.25 ± 3.29 a			
*Sclerotinia sclerotiorum*	100	26.06 ± 2.24 b	y = 2.468x − 1.8093	R^2^ = 0.9583	574.16
	300	38.97 ± 2.83 a			
	500	38.97 ± 3.29 a			
	1000	43.23 ± 3.17 a			
	2000	43.28 ± 4.29 a			
*Gibberella sanbinetti*	100	29.21 ± 2.13 d	y = 0.4513x + 3.5151	R^2^ = 0.9207	1951.07
	300	55.93 ± 6.83 c			
	500	70.59 ± 6.28 b			
	1000	83.35 ± 7.81 a			
	2000	91.23 ± 5.28 a			
*Phytophthora infestans*	100	3.44 ± 0.59 d	y = 1.4789x + 1.5069	R^2^ = 0.9979	430.12
	300	18.44 ± 2.33 c			
	500	32.25 ± 2.39 b			
	1000	52.75 ± 4.74 a			
	2000	53.75 ± 4.29 a			
*Colletotrichum lagenarium*	100	29.21 ± 2.13 d	y = 1.5575x + 0.1975	R^2^ = 0.9454	1211.90
	300	55.93 ± 6.83 c			
	500	70.59 ± 6.28 b			
	1000	83.35 ± 7.81 a			
	2000	91.23 ± 5.28 a			
*Valsa mali*	100	2.22 ± 0.38 e	y = 4.1847x − 5.6559	R^2^ = 0.9427	351.88
	300	27.22 ± 1.38 d			
	500	80.13 ± 6.79 c			
	1000	87.97 ± 5.32 b			
	2000	100 ± 0.00 a			
*Alteraria alternata*	100	10.3 ± 1.38 c	y = 2.4598x − 1.3359	R^2^ = 0.9304	376.51
	300	26.47 ± 3.29 b			
	500	75.29 ± 6.27 a			
	1000	79.41 ± 5.38 a			
	2000	83.17 ± 6.29 a			
*Helminthosporium maydis*	100	2.91 ± 0.52 d	y = 4.1028x − 5.8397	R^2^ = 0.8485	438.56
	300	7.27 ± 0.81 d			
	500	55.75 ± 4.29 c			
	1000	71.93 ± 3.89 b			
	2000	100 ± 0.00 a			
*Phomopsis asparagi*	100	3.29 ± 0.43 a	y = 1.0702x + 1.2992	R^2^ = 0.8164	2871.08
	300	22.63 ± 2.11 a			
	500	28.03 ± 1.39 a			
	1000	28.68 ± 1.93 a			
	2000	36.45 ± 1.29 a			

Note: Values followed by different lower case letters indicate significant differences (*p* < 0.05) among treatments within each column according to the least significant difference (LSD) test.

**Table 2 microorganisms-10-00971-t002:** Inhibitory effect of wuyiencin and ε-PL at different concentrations on the mycelial growth of *Botrytis cinerea*.

Concentration (µg/mL)		Colony Diameter (cm)	Inhibition Rate (%)
ε-PL	Wuyiencin		
0	0	7.73 ± 0.64 d	-
	12.5	6.62 ± 0.55 c	14.36 ± 7.14 c
	25	4.73 ± 0.39 b	38.81 ± 5.10 b
	50	0.92 ± 0.07 a	88.10 ± 1.00 a
100	0	7.70 ± 0.64 d	0.39 ± 0.22 d
	12.5	6.07 ± 0.51 c	21.47 ± 3.78 c
	25	3.80 ± 0.32 b	50.84 ± 2.37 b
	50	0.92 ± 0.08 a	88.10 ± 0.57 a
200	0	6.20 ± 0.52 c	19.79 ± 6.68 c
	12.5	6.05 ± 0.50 c	21.73 ± 6.52 c
	25	3.47 ± 0.29 b	55.50 ± 3.74 b
	50	0.37 ± 0.17 a	95.21 ± 0.40 a
300	0	5.33 ± 0.70 d	31.05 ± 5.75 d
	12.5	4.08 ± 0.20 c	47.22 ± 4.40 c
	25	2.07 ± 0.10 b	73.22 ± 2.23 b
	50	0.30 ± 0.15 a	96.12 ± 3.23 a
400	0	3.20 ± 0.27 d	58.60 ± 3.45 d
	12.5	2.78 ± 0.23 c	64.04 ± 3.00 c
	25	1.32 ± 0.11 b	82.92 ± 1.42 b
	50	0.18 ± 0.02 a	97.67 ± 0.19 a
500	0	2.23 ± 0.19 c	71.15 ± 2.40
	12.5	2.02 ± 0.10 b	73.87 ± 2.18 c
	25	0.95 ± 0.79 a	87.71 ± 5.91 b
	50	0.00 ± 0.00 a	100 ± 0.00 a

Note: Colony diameters were measured 5 days after inoculation. Values followed by different lower case letters indicate significant differences (*p*< 0.05) among treatments within each column according to the least significant difference (LSD) test.

**Table 3 microorganisms-10-00971-t003:** Inhibitory effect of wuyiencin and ε-PL at different concentrations on conidial germination of *Botrytis cinerea*.

Treatment	Concentration (μg/mL)	Germination Rate (%)	Inhibition Rate (%)
Control	-	51.00 ± 2.50	-
ε-PL	50	47.00 ± 3.00 f	7.84 ± 3.39 f
	100	26.00 ± 2.00 e	49.02 ± 3.92 e
	200	21.00 ± 2.00 d	58.82 ± 3.92 d
	300	15.00 ± 1.00 c	70.59 ± 1.96 c
	400	10.00 ± 1.50 b	80.39 ± 2.94 b
	500	4.00 ± 1.00 a	92.16 ± 1.96 a
wuyiencin	10	42.00 ± 2.00 f	17.65 ± 3.92 f
	20	35.00 ± 2.50 ef	31.37 ± 4.90 e
	40	30.00 ± 3.00 de	41.18 ± 5.89 d
	60	12.00 ± 1.00 cd	76.47 ± 2.94 c
	80	8.00 ± 1.50 bc	84.31 ± 1.96 b
	120	2.00 ± 0.50 ab	96.08 ± 0.98 a
ε-PL + wuyiencin	200 + 10	17.50 ± 1.00 a	65.68 ± 1.96 d
	200 + 20	11.00 ± 1.50 c	80.00 ± 2.94 c
	200 + 40	6.00 ± 1.00 b	88.23 ± 1.96 b
	200 + 60	0.00 ± 0.00 a	100 ± 0.00 a
	200 + 80	0.00 ± 0.00 a	100 ± 0.00 a
	200 + 120	0.00 ± 0.00 a	100 ± 0.00 a

Note: Values followed by different lower case letters indicate significant differences (*p* < 0.05) among treatments within each column according to the least significant difference (LSD) test.

**Table 4 microorganisms-10-00971-t004:** Control efficiencies of wuyiencin and ε-PL on tomato grey mold in the leaves.

Treatment (μg/mL)	Colony Diameter (cm)	Inhibition Rate(%)
Control	1.20 ± 0.20 a	-
Wuyiencin 60	0.73 ± 0.07 b	38.89 ± 6.11 c
ε-PL 1500	0.33 ± 0.05 c	72.22 ± 4.17 b
Wuyiencin 60 + ε-PL 1500	0.13 ± 0.02 d	88.89 ± 1.67 a
ε-PL 2000	0.46 ± 0.06 cd	61.67 ± 10.93 ab
Wuyiencin 60 + ε-PL 2000	0.20 ± 0.03 cd	83.33 ± 1.67 ab
ε-PL 2500	0.60 ± 0.00 b	50.00 ± 0.00 c
Wuyiencin 60 + ε-PL 2500	0.67 ± 0.05 b	44.44 ± 4.17 c
ε-PL 3000	0.65 ± 0.04 b	45.44 ± 3.33 c
Wuyiencin 60 + ε-PL 3000	0.63 ± 0.10 b	47.78 ± 0.83 c

Note: Values followed by different lower case letters indicate signifificant differences (*p* < 0.05) among treatments within each column according to a least significant difference (LSD) test.

**Table 5 microorganisms-10-00971-t005:** The FICI of ε-PL combined with wuyiencin against *B.cinerea*.

Treatment	MIC(μg/mL)	FIC		FICI
		Wuyiencin	ε-PL	
Wuyiencin (alone)	40	0.13	0.33	0.46
ε-PL (alone)	30			
Wuyiencin (in combination)	5			
ε-PL (in combination)	10			

## Data Availability

Data are contained in this manuscript or Appendix A.

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
