# Peer review of "Effects of ε-Poly-L-Lysine Combined with Wuyiencin as a Bio-Fungicide against Botryris cinerea"

_microorganisms, 2022, doi:10.3390/microorganisms10050971_

Round 1

Reviewer 1 Report

Summary

Authors well presented the broad fungicidal activity of ε-poly L lysine through in vitro experiments and focused on the activity against B. cinerea including the results about the combination of ε-poly L lysine with a second bio-fungicide, wuyiencin. They demonstrated for the combination ε-poly L lysine + wuyiencin the inhibition of B. cinerea spores germination as well as the negative effect on the mycelium morphology and integrity. Furthermore through tests on tomato leaves the efficacy in vivo against the pathogen was confirmed. ε-poly L lysine and wuyiencin seems to be also involved in tomato defense responses (increased defense related enzymes activity).

General comments 

In general the experiments have been properly planned and executed and results are in most of the cases clearly reported in charts, tables and figures. The paper add some interesting new results to the already known information about ε-poly L lysine and wuyiencin antifungal activity.

Despite that I found quite a lot of typos, repetitions of words within the sentences, forgotten words, inconsistency in text formatting. A more careful revision of the text would have been appropriate. In few cases I also detected inconsistency between results reported in discussion and results sections.

Furthermore in my opinion one of the main statement done in the paper, the synergy of the combination ε-poly L lysine + wuyiencin should be better demonstrated with appropriate formulas and calculations (see details below in specific comments section).

Specific comments 

Between " quotation marks " I reported text from the manuscript.

Page 2, last paragraph of Introduction. "We revealed that e-PL combined with wuyiencin had synergistic effects against B. cinerea..."

In the manuscript it is not clearly demonstrated that e-PL combined with wuyiencin has synergistic effects. This should be supported by dedicated formulas to define if the observed efficacy level of the combination overcome the synergy threshold. I think it is important to prove to the reader that the synergy is present. You could use the Colby's formula or other approaches to clearly show that the combination has a synergistic effect.
In one paper, cited in the manuscript [6],  the synergy between nisin A and e-poly-L-lysine was demonstrated and authors proved the synergy concept with formulas and dedicated explanations using the fractional inhibitory concentration index (FIC). Probably synergy is present in your data but you have to prove it with the proper calculations in my opinion. 

Section 2.2, first sentence. All the twelve scientific names should be written in italics.

Section 2.2, last sentence. "Tomato plants were grown in the artificial greenhouse at a constant temperature of 28 °C" Please, add also information about other relevant parameters like Relative Humidity and photoperiod.

Section 2.3,"At day 5 days post inoculation (dpi) the mean diameters of the colonies were determined."

Section 2.4 "Different solutions were mixed with PDA medium and medium lacking both e-PL and wuyiencin solution was used as a control as control."

Section 2.6 "The spore germination rate was calculated was calculated for 200 spores each time."

Section 2.7 "Tomato seeds were grown under the conditions of a 12-hour light-dark cycle for at 26°C." Later in the same section "The assays were carried out three times independently."

Section 2.7 "Control leaves were sprayed using distilled water. Twenty-four hours later, the leaves were sprayed with a 2% glucose solution, inoculated with B. cinerea, and cultured in an incubator with 90% humidity with daily water spraying to retain moisture. After 3 days, the diameter of disease spots on leaves were determined and the disease incidence was investigated. Twenty leaves were inoculated with each concentration. The assays were carried out three times independently. Percent inhibition (PI) = (dc-dt )/dc × 100, where dc represents the diameter of the control (untreated) colony and dt represents the diameter of the treated colonies."

I can not fully understand the disease assessment methods. Colony diameter of all disease spots on leaves (per object) is measured and the average value is reported in results (Results in Table 4) ? If this is the case also the number of disease spots considered to determine the average diameter of the spots should be mentioned. Please, describe better these aspects.

Section 3.1 First sentence. Scientific names should be in italics

Section 3.1 "With the lower concentration of 500 µg/mL e-PL, the inhibition rate against B. cinerea, Sclerotinia sclerotiorum, Colletotrichum lagenarium, Valsa mali and Alteraria alternata was above 70%." At 500 µg/mL e-PL the inhibition rate against Sclerotinia sclerotiorum was equal to 38,97%, far lower than 70%. Please, delete Sclerotinia sclerotiorum from this sentence.

Table 1. Note: "Values followed by different lower case letters indicate significant differences (P < 0.05) amongtreatments within each column according to the least significant difference (LSD) test." Change in among treatments.

Table 3. In Note there is the same typo reported for Table 1 (see above)

Section 3.5 "By contrast, with increasing e-PL concentration, the control effect decreased and the leaves appeared chlorotic and had black spots (Figure 5)." This result it was correctly reported in results section 3.5 but it has not been described or explained in Discussion section.

Section 3.5 "When wuyiencin was used alone at 60 µg/mL, the inhibition rate was only 38.89%." Where I can find in the manuscript this result about Wuyiencin 60 µg/mL showing inhibition rate of 38.89% ?

Section 3.6 title. "3.6. Scanning electron microscope observation of the effects of combined treatment on the pathogen on isolated tomato leaves under"

Section 3.6 "Treatment with the combination therapy resulted in the
lowest number of mycelia,..." Please, rephrase the sentence to better described the results. I would mention mycelia density or something similar..

Figure 6. Caption B. cinerea should be in italics

Figure 7. Caption "The effect of wuyiencin and e-PL on defense-related enzyme activities in tomatoes. 50 mg/L wuyiencin, 1500 µg/mL e-PL, 50 mg/L wuyiencin + 1500 µg/mL e-PL and sterile water (control) were applied 3 days before the inoculation of B. cinerea spores (107cfu/mL)." 50 mg/L wuyiencin is reported in caption of figure 7 while in the figure itself I can read 60 mg/L. Which concentration is the correct one ?

Discussion 4. "e-PL at 200 µg/mL had an inhibitory rate toward mycelial growth of 83.9% against A. alternata [13]." The result reported here from literature is quite different from what observed in your experiment. Table 1 shows for e-PL 300 µg/mL an inhibition rate of 26,47% against A. alternata.
Can you explain or propose an hypothesis for such difference ?

Discussion 4. "e-PL at 1200 µg/mL reduced the lesion are by 76.28%, and..."

Discussion 4 "In addition, combinations of e-PL with other compounds, such as endolysin,..."

Discussion 4 "Similarly, combined treatment with e-PL at 200 µg/mL + wuyiencin at 80 µg/mL completely inhibited spore germination (Figure 4)." According to Figure 4 wuyiencin was tested at 60 µg/mL. Please, double check.

Discussion 4. "In this study, we observed significantly increased defense-related enzyme activity after treatment with the combination of wuyiencin and lysine compared with that e-PL induced by either agent alone." Where I can see that the differences of the enzyme activity observed between combination of wuyiencin and e-PL in comparison to the enzyme activity of each agent alone are statistically significant ? In Figure 7, I can see error bars but it is not clear if a statistical analysis of the data have been performed. Please, clarify.

Discussion 4. "Previous in vitro leaf tests showed that high concentrations
of e-PL had certain harmful effects on tomato plants. However, when mixed with wuyiencin, not only did the prevention effect improved, but also the damage was significantly reduced (Figure 4), which is consistent with the results of previous field trials of e- PL on strawberries (Figure S2)." Looking at Figure S2 I can see that the addition of wuyiencin to e-PL seems to led to higher damages on strawberry leaves. it seems the opposite to what stated in the manuscript text.

Discussion 4. "The concentration most suitable for inhibiting spore germination was 200 µg/mL e-PL + 80 µg/mL wuyiencin." Observing results (check Table 3)  e-PL 200 µg/mL + wuyiencin 60 µg/mL already showed 100% spore germination inhibition therefore this could be the most suitable concentration to suggest.

Discussion 4. "However, whether the combined treatment can control B. cinerea infection of tomatoes and it mechanism of action require field tests and further experiments."  First of all the sentence should be adjusted and improved from the English point of view. Furthermore I think this final sentence goes to far... the efficacy of the bio-fungicides combination has been proved in vitro tests and on detached tomato leaves and the sentence should reflect these results and not be so generic (..control B. cinerea infections of tomatoes..). I would add also another consideration: there are several important steps of development to follow other than the characterization of the MoA to get closer to the use in crop protection of this combination of bio-fungicides. For example: compatibility of the bio-fungicides (chemical-physical compatibility of the two bio-fungicides); feasibility of the industrial scale-up; improvement of the rainfastness; clear evaluation about possible phytotoxicity issues; efficacy tests using patho-systems closer to real agricultural practice starting from experiments under greenhouse conditions.

Author Response

Thanks for reviewer's suggesion. According to the reviewers’ comment, the manuscript has been revised and uploaded  PDF in the system. 

Reviewer 2 Report

Dear colleagues. There are questions and comments, they are in the attached file. 

Author Response

Response to Reviewer: 2

  1. Page 2. “its antibacterial activity and low toxicity» - Toxicity to which organisms do the authors have in mind?

Our response: Thanks for reviewer’s careful checking. After reference checking,ε-PL is non-toxicity.

  1. “Twelve pathogenic fungi were used: Physalospora piricola, Rhizopus stolonifer, Botrytis cinerea, Fulvia fulva, Sclerotinia sclerotiorum, Gibberella sanbinetti, Colletotrichum lagenarium, Physalospora piricola, Valsa mali, Alteraria alternata, Helminthosporium maydis, and Phomopsis asparagi”

- The names are repeated twice. In the following, Phytophthora infestance is used, which is mentioned in Figure 1 on page 6.

Our response: Done as suggested.

  1. “These fungi were grown at 4 ℃ on potato dextrose agar (PDA) medium. Prior to the antagonist experiments, the fungi were activated at 28 ℃ for 5–7 day to determine their mycelial growth rates.“Pathogenic fungi were grown at 4℃? At this temperature, they grow poorly.

Our response: Thanks for reviewer’s careful checking.We have modified the sentence “These fungi were stored at 4 ℃ on potato dextrose agar (PDA) medium.”

  1. “Tomato plants were grown in the artificial -redundant word greenhouse at a constant temperature of 28 ℃”.

- What was the temperature regime and what kind of soil was used?

Our response: The information of the temperature regime and the soil was added.

  1. Page 4.“Twenty-four hours later, the leaves were sprayed with a 2% glucose solution, inoculated with B.cinerea, and cultured in an incubator with 90% humidity with daily water spraying to retain moisture.After 3 days, the diameter of disease spots on leaves were determined and the disease incidence wasinvestigated”.

   --Why was glucose added to the inoculum? What was the light mode? Did the authors try to infect the leaves of intact plants?

   Our response: 2% glucose solution could improve the inoculation efficient of B.cinerea. The information of light mode was added. Tomato plants could be a systemic infection by Botrytis cinerea.

  1. Page 6.It is necessary to correct the names. At what temperature was phytophthora cultivated?

Our response:  It has modified the name. The fungi were activated at 25℃.

  1. Page 14.Did the authors study only uninoculated plants? Have attempts been made to test the activation ofoxygen by plants?

Our response: It was used by inoculated plants. More detail information were added to the sentence “different bio-fungicides and distilled water were applied 3 days prior to B. cinerea spores inoculation. We compared tomato plants after treatment using distilled water (control), wuyiencin at 60 μg/mL, ε-PL at 1500 μg/mL, and wuyiencin at 60 µg/mL + ε-Pl at 1500 µg/mL.”

Thanks for reviewer’s suggestion. Antioxidant enzyme system was assessed in the study. It is a pity that we haven’t design the activation of oxygen by plants in this paper. We will consider to design the experiment in future study.

Round 2

Reviewer 2 Report

Dear colleagues, thank you for the answer, now it is more clear